# Analysis of Aluminum Oxides Submicron Particle Agglomeration in Polymethyl Methacrylate Composites

**DOI:** 10.3390/ijms24032515

**Published:** 2023-01-28

**Authors:** Vladimir Kuklin, Sergey Karandashov, Elena Bobina, Sergey Drobyshev, Anna Smirnova, Oleg Morozov, Maxim Danilaev

**Affiliations:** 1Department of Electronic and Quantum Information Transmission Devices, Kazan National Research Technical University n.a. A.N. Tupolev-KAI, Karl Marx St., 10, Tatarstan, 420111 Kazan, Russia; 2Radiophisics Department, Kazan Volga-Region Federal University, Kremlevskaya St., 18, Tatarstan, 420018 Kazan, Russia

**Keywords:** composites, particles agglomeration, optical performance, polymer composite, distributed reinforcement particles

## Abstract

Agglomeration of distributed particles is the main problem in polymer composites reinforced with such particles. It leads to a decrease in mechanical performance and its poor reproducibility. Thus, development of methods to address the agglomeration of particles is relevant. Evaluation of the size and concentration of agglomerates is required to select a method to address agglomeration. The paper analyzes aluminum oxide particles agglomeration in particles-reinforced polymethyl methacrylate (PMMA) composites. Quantitative parameters of polystyrene-coated aluminum oxide particles agglomerates are obtained for the first time in this article. Unlike uncoated aluminum oxide particles, when coated aluminum oxide particles are used, agglomerates concentration in polymer composites decreases approx. 10 times. It demonstrates that modification of submicron particles by a polymer coating decreases the number of agglomerates in the polymer composite. The use of transmittance and opacity values to estimate particles agglomerates is reasonable in this article. It is shown that the difference in optical performance of specimens reinforced with coated and the original particles is related to the number and average size of agglomerates in the specimens. For example, when the concentration exceeds 0.2%, transmittance values for the specimens reinforced with coated particles are greater than the ones for the specimens reinforced with the original particles.

## 1. Introduction

One of the challenges that polymer composites reinforced with distributed submicron particles face is agglomeration of these particles. It is well-known that agglomeration of particles affects substantially mechanical performance of such polymer composites [1,2]. For example, [3] demonstrates that a change in mechanical performance of ceramic composites with graphene is related to agglomeration radius and pore formation around these agglomerates. Among the main causes of submicron particles agglomerates formation are poor wettability of the particles in the polymer [4], a process used for composite production [5], and concentration of submicron particles in a composite [6].

It is possible to improve wettability of submicron particles in a polymer by modification of their surface, such as plasma treatment [7]; adding functional groups to the surface of the particles that change their wettability and can form chemical bonds with matrix polymer [8] or creating a polymer coating on the surface of the particles that ensures adhesion of matrix polymer to these particles [9,10]. It should be noted that it is impossible to get rid of the agglomerates completely even after modification of the submicron particles [11].

In the majority of studies there is no quantitative assessment of the agglomerates, including their number, average size, and concentration. For example, paper [10] analyzes how thickness of polymer coating on the surface of reinforcement particles impacts mechanical performance of a composite. Papers [12,13,14,15,16] estimate an impact of distributed particles concentration on mechanical performance of various composites. However, there are no data on the proportion of these particles’ agglomerates in the composite. Papers [17,18] should be mentioned as well. They study the effect of agglomerates on mechanical performance of polymer composites; however, they do not discuss agglomeration mechanism. So, it is impossible to find the most reasonable ways to reduce the number of agglomerates in polymer composites and formalize the challenge of production process optimization. Paper [19] suggests a method for analyzing the number and average size of agglomerates based on the results of processing photographs taken through electron microscopes. However, all experimental studies conducted to verify the suggested approach involve solutions where the kinetics of agglomerate formation is different compared to the viscous melt of the matrix polymer. Paper [20] also presents the studies of agglomerates in liquid dispersion medium: it is demonstrated that the size of Al_2_O_3_ agglomerates is reduced as nickel ions concentration in electrolyte decreases while there are no estimates of the effect that particles concentration has on the result. Optical methods used to study agglomerate properties in polymer composites need a special mention [21]. For example, paper [22] reviews the effect of silver nanoparticles agglomerates size on the optical performance of thin films: an increase in size leads to a dramatic decrease in reflectance and transmittance of thin polymer films. As it is required to solve an inverse problem, i.e., to define agglomerate parameters (their size and number) in this case, the results have low reliability because there is no unique solution of such a problem, especially taking into account measurement error [23].

The goal of this study is to discover how conformal coating of particles influences the parameters of agglomerates in particle-reinforced polymethyl methacrylate composites.

It is possible to determine agglomerates parameters over the entire volume of a polymer composite using optical microscopy, for example. This method provides a possibility to determine the dimensions of large agglomerates (1 micron and up) in optically transparent polymers. That is why this paper uses polymethyl methacrylate as a transparent matrix. Al_2_O_3_ submicron particles are originally spheres [9,10,24], and polystyrene coating changes their geometry. Due to this fact electron microscopy makes it possible to see if the coating is present on the particles.

## 2. Results

Figure 1 shows results of statistical analysis of specimens’ microphotographs: polymethyl methacrylate reinforced by original and coated submicron Al_2_O_3_ particles (Table 1). The figure has the following notations: *D*_0_—average agglomerate diameter. For the samples with original particles *D*_max_ = 14 micron, with coated particles *D*_max_ = 13 micron.

Images obtained using scanning electron microscope were analyzed to study wettability of submicron particles (original and coated). Figure 2 shows the examples of images. These are the images of the chips on specimens 1.3 and 2.3 (Table 1). It should be noted that coated particles are difficult to locate on the surface when obtaining images at secondary emission (Figure 3): particles in the chip area are coated with the polymer. Analysis of the images after secondary emission provides a possibility to distinguish surface areas with different molecular mass [25]. It is the analysis of these images (Figure 2c), e.g., the chip on specimen 2.3 (Table 1), that confirms better interaction of coated particles with the polymer than the original ones [10].

Figure 3 shows an example of agglomerates size distribution histogram for specimen 2.2. Figure 4 demonstrates an example of a micrograph for specimen 2.2. The number of submicron particles agglomerates was normalized to 230.

Figure 5 shows the changes in optical performance of composite specimens; these results were obtained by photometric method [26]. The figure has the following notations: *M*_K_, *K*_K_—opacity and transmittance of the specimens reinforced with coated particles, *M*_H_, *K*_H_—opacity and transmittance of the specimens reinforced with the original particles.

## 3. Discussion

Agglomeration takes place when two processes run at the same time:A decrease in the number and size of agglomerates due to an increase in PMMA melt viscosity while particles concentration increases;An increase in the number and size of agglomerates while particles concentration increases due to an increase in the number of particle collisions, including inelastic collisions [27].

When particles concentration grows, the distance between them decreases, the frequency of inelastic collisions increases, respectively. This leads to an increase in the number of agglomerates in the specimens reinforced with the original (non-coated) particles. Together with an increase in concentration, material viscosity increases [28], and when the viscosity is high, particle mobility decreases as well as the frequency of collisions [29]. Apparently, the first process prevails in specimens reinforced with the original particles; that explains the increase in agglomerates concentration (Figure 6a).

The size of the particles and the quality of their surface determine the likelihood of inelastic collisions [30]. We believe that inelastic collisions of coated particles are more likely than of the original ones. The reason behind that is low molecular mass of polystyrene on the surface of the coated particles. Low molecular polymers have high plasticity [31]. So, when coated particles collide, a part of kinetic energy transfers into deformation of the coating which leads to an increased probability of inelastic collisions [30].

Due to a better adhesion of polymer matrix to coated particles, their drag in the polymer melt increases compared to the original particles [30]. It is an additional reason behind the decrease in the frequency of coated particles collisions in the melt, illustrated by the curve steadily going downwards in Figure 1a. It should be noted that the dimensions of coated and the original agglomerates in polymer composite specimens are close (Figure 1b): the frequency of collisions between the original particles is higher than between the coated ones and inelastic collisions between coated particles are more likely than between the original ones.

Optical performance of the specimens with the same mass concentration of coated and original particles in an ideal case (when there are no agglomerates) has to be the same [32]. It is explained by the little thickness of the polystyrene coating on the surface of these particles [10,33]. So, different optical performance of the two types of specimens may be explained only by agglomerates of submicron particles with different parameters (concentration and average size).

Agglomerates fraction in the specimens compared to distributed submicron particles and the average size of agglomerates govern opacity and transmittance [34]. Specimens reinforced with coated particles demonstrate better transmittance than specimens reinforced with the original particles when the concentration of the particles exceeds 0.2%. This happens because starting from 0.2% agglomerates concentration in specimens 2.1–2.3 (Table 1) is substantially lower than in specimens 1.1–1.3 while the average size of agglomerates is virtually the same. At lower concentrations (up to 0.2%), agglomerates in the specimens reinforced with coated particles are approx. twice as large as the ones in the specimens reinforced with the original particles. That is why, apparently, the transmittance of specimen 1.4 is lower and opacity is higher compared to the specimen 2.4.

## 4. Materials and Methods

### 4.1. Materials and Their Properties

Polymer composite specimens with submicron particles mass fraction of up to approx. 1% were studied. Submicron aluminum oxide (Al_2_O_3_) particles were used: Al_2_O_3_ (a mixture of δ- and θ-phases in a size range of 40/190 nm) submicron particles were produced by Plasmotherm (Product number: PL1344281). Optically transparent polymethyl methacrylate (SP-2) was used as a polymer matrix. Aluminum oxide particles were modified by application of a polystyrene coating on their surface using vapor phase method, described in [10].

Coated particles were deposited on polymethyl methacrylate (PMMA) powder while stirring to obtain uniform distribution of coated particles in PMMA. For that purpose, a powder of coated particles was gradually added to PMMA powder while stirring using ITA-07 powder mixer (Italy). Mixing speed was 20 ± 5 rpm. After mixing, the specimens of PMMA composite were obtained within 30 ± 5 min. Specimens of Al_2_O_3_ particles-reinforced PMMA composite (both the original ones and the ones coated with polystyrene) were produced by injection molding at the following conditions: solution temperature 190 ± 5 °C; mold temperature 200 ± 5 °C; time under pressure of 41.6 MPa (424.4 kg-force/cm^2^) 10 min; cooling time 60 ± 10 min. The specimens were cooled down naturally at an ambient temperature of 24 ± 2 °C. Four types of specimens with different mass concentration of particles were produced for experimental study of agglomerates (Table 1). Specimens’ dimensions were 120 × 20 × 3 mm.

### 4.2. Methods

#### 4.2.1. Al_2_O_3_ Particles Coating

Aluminum oxide particles were modified by application of a polystyrene coating on their surface using vapor phase method, described in [9,10,24]. The main idea behind this method is to generate a two-phase gas flow of Al_2_O_3_ particles agglomerates, their dispersion in corona and then mixing of these particles with styrene vapor. The coating is generated due to condensation of styrene on the surface of the particles. It should be noted that polystyrene in the coating has low molecular mass of approx. 40,000 Da [24]. The coating on the surface of the particles was controlled by scanning electron microscopy (Figure 1).

A study of adhesion between polystyrene coating and aluminum oxide particles [9] demonstrated covalent bond of polystyrene molecules to surface atoms of these particles. This bond keeps the coating on the surface of the particles when particle-reinforced PMMA specimens are molded.

#### 4.2.2. Analysis of the Agglomerates

The size of agglomerates and their number were determined by optical microscopy followed by statistical analysis of micrographs in ImageJ software. Micrographs were taken in a few sections along the specimen. Changing focal points allowed taking a few through-thickness micrographs. Only those agglomerates that were located in focal points and had distinct outline were recorded using Adobe Illustrator software. Statistical analysis of the images was performed in ImageJ software and agglomerates size distribution histograms were plotted. Micromed microscope with FMA050 Touptek Photonics adapter with 40× *g* magnification made it possible to measure approx. 1 ± 0.1 micron agglomerates.

#### 4.2.3. A Method for Measurement of Optical Properties

This paper compares agglomerates analysis results (concentration and average size) with optical performance of the specimens (transmittance and opacity). Optical performance was measured using turbidimetric photometer, described in [26]. BL-L101URC LEDs with the wavelength of 660 nm provided the source of light. These LEDs were powered by 1522 Hz impulse voltage with 15–20 mA current stabilization. Output voltage from measuring and reference photodiodes was amplified and registered by dual-trace digital storage oscilloscope with an 8-bit ADC and 100 kHz sampling rate, and each 10,000 readings were recorded in a file.

## 5. Conclusions

The assessed concentration and average size of aluminum oxide submicron particles agglomerates in polymethyl methacrylate confirmed that modification of the submicron particles provides a possibility to reduce the number of agglomerates in polymer composites. Only large-scale agglomerates (exceeding 1 micron) were analyzed, so optical microscopy could be used to obtain the images of agglomerates in optically transparent PMMA specimens and statistical analysis of these images could be performed. Unlike uncoated aluminum oxide particles, when coated aluminum oxide particles are used, agglomerates concentration in polymer composites decreases approx. 10 times. Moreover, when particles concentration in the specimens increases, the ratio of agglomerates concentration in the specimens reinforced with the original particles and in the specimens with coated particles increases as well. It is demonstrated that modification of submicron particles by polymer coating reduces the number of agglomerates in a polymer composite. The paper highlights that parameters of the agglomerates depend on the frequency of inelastic collisions of distributed particles in a polymer melt during composite specimens molding. At the same time, when concentration increases, the average distance between particles distributed in a polymer melt decreases, and that leads to an increase in the frequency of inelastic collisions. On the other hand, an increase in particles concentration contributes to an increase in melt viscosity that reduces particle mobility and therefore leads to a decrease in the frequency of inelastic collisions between the reinforcement particles. Moreover, an increase in adhesion between the submicron particles and matrix polymer promotes an increase in drag of these particles in the melt that additionally decreases the frequency of inelastic collisions between the particles and leads to a lesser agglomeration of such particles. Optical performance of the specimens, such as transmittance and opacity, may be used to assess agglomerate fraction in polymer composite specimens with optically transparent polymer matrix. When the concentration exceeds 0.2%, transmittance values for the specimens reinforced with coated particles are greater than the ones for the specimens reinforced with the original particles. When the concentration is below 0.2%, the size of agglomerates in the specimens reinforced with coated particles is approx. twice as large as in the specimens reinforced with the original particles. A suggested method for determining the parameters of submicron particles agglomerates in optically transparent polymers (polymethyl methacrylate, polycarbonate) may be applied for various particles and surface modification techniques.

## Figures and Tables

**Figure 1 ijms-24-02515-f001:**
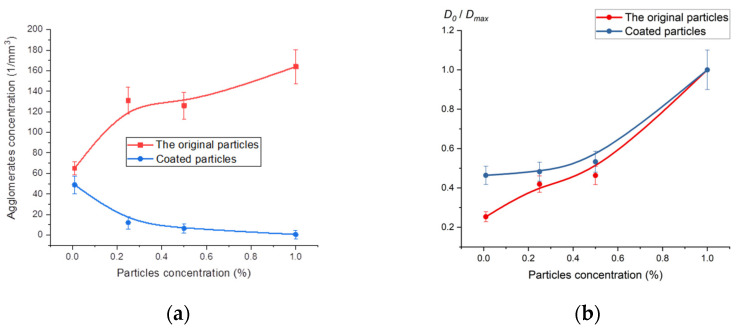
(**a**) Agglomerates concentration *N* vs. mass concentration of submicron reinforcement particles in the polymer composite; (**b**) Scaled average diameter of particles *D*_0_/*D*_max_ vs mass concentration of submicron reinforcement particles in the polymer composite.

**Figure 2 ijms-24-02515-f002:**
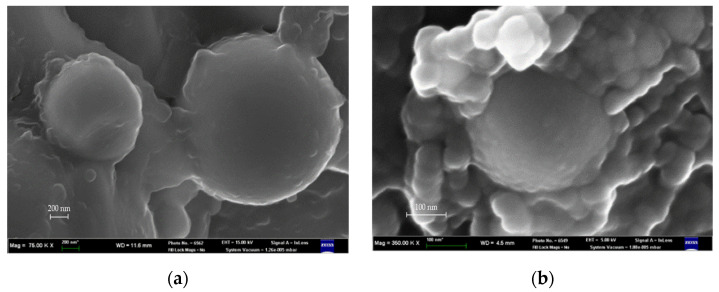
SEM-images of Al_2_O_3_-reinforced composite specimens: (**a**) the original particles, (**b**,**c**) coated particles.

**Figure 3 ijms-24-02515-f003:**
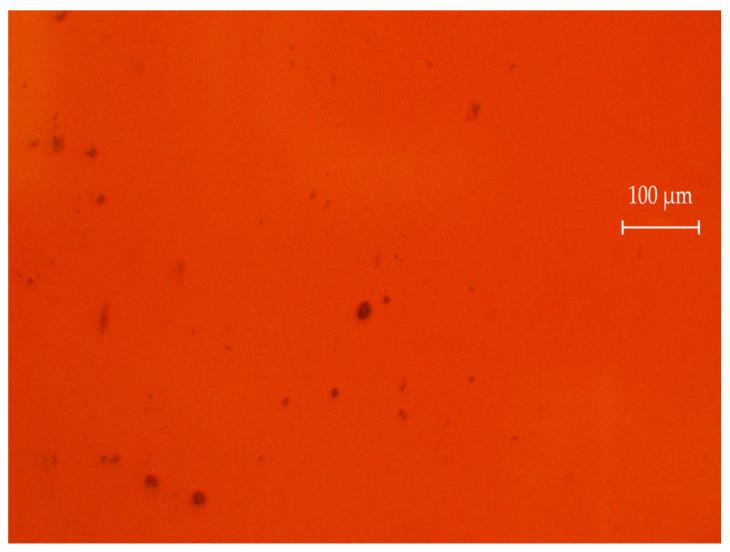
A typical micrograph of an optical image of specimen 2.2.

**Figure 4 ijms-24-02515-f004:**
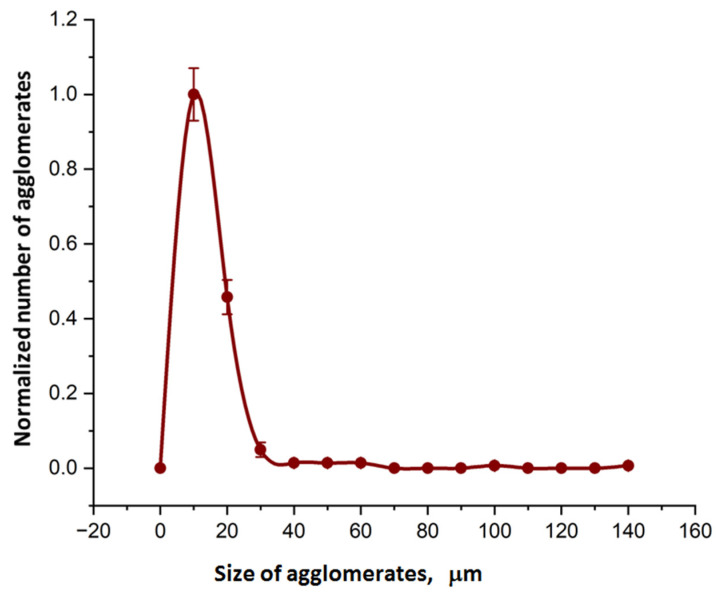
Agglomerates size distribution histogram for specimen 2.2.

**Figure 5 ijms-24-02515-f005:**
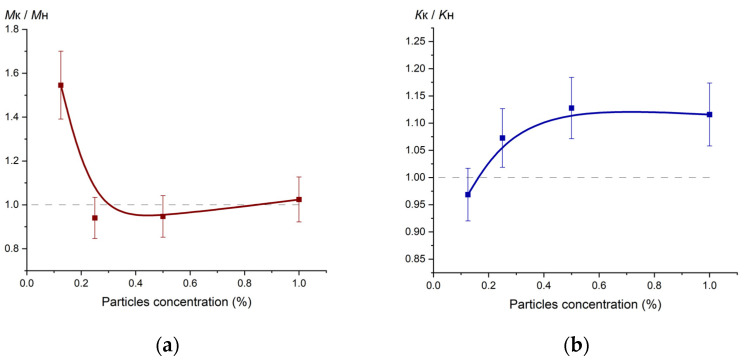
Optical performance of the polymer composite: (**a**) opacity vs. particles concentration; (**b**) transmittance vs particles concentration.

**Figure 6 ijms-24-02515-f006:**
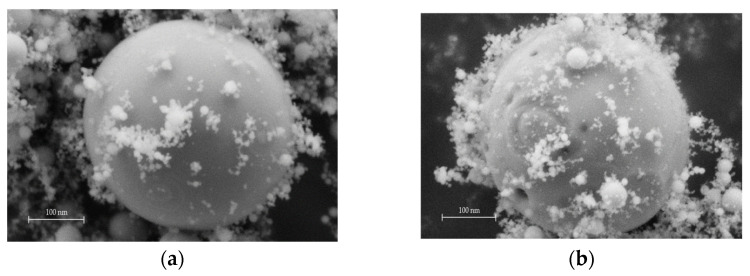
Microphotographs of aluminum oxide particles: (**a**) the original particles; (**b**) coated particles.

**Table 1 ijms-24-02515-t001:** Classification of the specimens.

Specimen Type	Polystyrene Coating?	Mass Concentration of Particles in the Specimen, %
Specimen 0	No	0
Specimen 1.1	No	1
Specimen 1.2	No	0.5
Specimen 1.3	No	0.25
Specimen 1.4	No	0.01
Specimen 2.1	Yes	1
Specimen 2.2	Yes	0.5
Specimen 2.3	Yes	0.25
Specimen 2.4	Yes	0.01

Five specimens of each type were manufactured to improve the reliability of experimental data.

## Data Availability

Not applicable.

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
