# Peer review of "Analysis of Aluminum Oxides Submicron Particle Agglomeration in Polymethyl Methacrylate Composites"

_ijms, 2023, doi:10.3390/ijms24032515_

Round 1
Reviewer 1 Report
In this paper, the agglomeration characteristics of nano particles in polymer are studied, and the effects of collision frequency and viscosity on particle growth are analyzed. It is instructive for engineering and science. Due to the small amount of data, it is impossible to see the critical point of concentration effect on viscous and inelastic collisions.
Author Response
In this paper, the agglomeration characteristics of nano particles in polymer are studied, and the effects of collision frequency and viscosity on particle growth are analyzed. It is instructive for engineering and science. Due to the small amount of data, it is impossible to see the critical point of concentration effect on viscous and inelastic collisions.
Thank you for the remark. The authors did not aim at evaluating the concentration effect on the parameters of elastic and inelastic collisions of the particles in polymer. We agree with the reviewer that these studies are by far of interest and may be the subject of a separate article. The assumptions about agglomeration mechanisms, made by the authors, are based on the well-known principles of particles collision dynamics [30]. There is no point in theoretical estimation of elastic and inelastic collisions cross-section due to the lack of data about the impact factors.
Reviewer 2 Report
Title: Analysis of submicron particles agglomeration in polymethyl methacrylate composites.
##Overall comments
The paper describes the study on the formation of agglomeration of styrene-coated submicron Al2O3 particles in MMA. The paper is interesting. However, the weak point of the paper is that the formation of composites needed to be characterized in detail. Authors have used only surface morphological studies using SEM. The paper's novelty could be more sound because the authors have published some parts of this work in different journals. The paper may consider for publication after a Major revision.
##Comments on the title, abstract, and references
1. The title should be specific according to this study. The reviewer suggests a title, and authors may consider it. "Analysis of Aluminum oxides submicron particle agglomeration in polymethyl methacrylate composites.
2. The abstract does not focus on this study's novelty, results, and utility.
3. Keywords should be at least five.
4. Suggest including more references.
##Comments on Introduction
5. Line 36-37: In the majority of studies, there is no quantitative assessment of the agglomerates, including their number, average size, and concentration. Please add references.
6. Please discuss the novelty of the findings compared to reference 10.
7. Why did the authors choose MMA and Al2O3 for this study? Please include the clarification in the introduction section.
8. Are there other reports using the same concept besides Al2O3 and MMA?
##Comments on methodology
9. The materials section should be separated and outline the details of the materials used in this study.
10. Please add a polystyrene-coated Al2O3 and composite preparation section and include the flowing clarification.
11. What was the density of styrene vapor in the reaction chamber (2), and what was the gas pressure in the chamber (2)?
12. What was the turbidity and pH of the water? How have authors used UV irradiation?
13. Please provide a complete schematic of this synthesis process.
14. Line 74: 'The coating on the surface of the particles was controlled by scanning electron microscopy' How?
15. With scale, SEM micrographs are meaningful.
16. Line 78: 'A study of adhesion between polystyrene coating and aluminum oxide particles [9]. The reviewer can not access the paper; sorry for it. The reviewer wants to know the mechanism of the formation of covalent bonds between aluminum oxide particles and styrene molecules.
17. Lines 82-83: 'Coated particles were deposited on polymethyl methacrylate (PMMA) powder while stirring to get uniform distribution of coated particles in PMMA' How?
18. Lines 86-87: Please remove the hyphen from each condition, e.g., temperature – 200±5оС.
19. Please include the image of each specimen in table 1 or Fig. 1.
20. Please separate the characterization section.
21. Lines 95-96: What type of micrographs were used for analyzing agglomerates by ImageJ software? Are these SEM images? What was the scale? Have authors used the same scale images for each specimen? Please provide those micrographs in a figure.
22. Line 100: Please provide the histograms in Figure 2.
23. Therefore, Lines 58-112; need modifications.
##Comments on results and discussion
24. Please delete the comma and use a point in the scale of Fig. 2b.
25. The scale of magnification must be included in Fig. 3
26. Lines 134-139: It is in brief. Please a detail.
27. If possible, please add at least two characteristics of composites from FTIR, XRD, and TGA.
27. Please add a discussion section separately.
##Comments on conclusion
28. Some of the results and discussion are in the conclusion section. Please write the findings and limitations of this research, future scope, and utility.
Author Response
##Comments on the title, abstract, and references
- The title should be specific according to this study. The reviewer suggests a title, and authors may consider it. "Analysis of Aluminum oxides submicron particle agglomeration in polymethyl methacrylate composites.
The authors are grateful to the reviewer for the suggested title. The title of the article is changed to “Analysis of Aluminum oxides submicron particle agglomeration in polymethyl methacrylate composites”.
- The abstract does not focus on this study's novelty, results, and utility.
The abstract is revised based on the reviewer’s feedback as follows:
“Agglomeration of distributed particles is the main problem in polymer composites reinforced with such particles. It leads to a decrease in mechanical performance and its poor reproducibility. Thus, development of methods to address the agglomeration of particles is relevant. Evaluation of the size and concentration of agglomerates is required to select a method to address agglomeration. The paper analyzes aluminum oxide particles agglomeration in particles-reinforced polymethyl methacrylate (PMMA) composites. Quantitative parameters of polystyrene-coated aluminum oxide particles agglomerates are obtained for the first time in this article. Unlike uncoated aluminum oxide particles, when coated aluminum oxide particles are used, agglomerates concentration in polymer composites decreases approx. 10 times. It demonstrates that modification of submicron particles by a polymer coating decreases the number of agglomerates in the polymer composite. The use of transmittance and opacity values to estimate particles agglomerates is reasonable in this article. It is shown that the difference in optical performance of specimens reinforced with coated and the original particles is related to the number and average size of agglomerates in the specimens. For example, when the concentration exceeds 0.2%, transmittance values for the specimens reinforced with coated particles are greater than the ones for the specimens reinforced with the original particles.”
- Keywords should be at least five.
Keywords are added based on the reviewer’s feedback:
“composites; particles agglomeration; optical performance; polymer composite; distributed reinforcement particles”
- Suggest including more references.
Based on the reviewer’s suggestion, the following papers are added to the references:
- Revo, S.L.; Avramenko, T.G.; Melnichenko, M.M.; Ivanenko, K.O. Mechanical characteristics of nanocomposite materials based on polytrifluorochloroethylene. Molecular Crystals and Liquid Crystals 2022, 1-8. https://doi.org/10.1080/15421406.2022.2067663
- Dal Lago, E.; Cagnin, E.; Boaretti, C.; Roso, M.; Lorenzetti, A.; Modesti, M. Influence of different carbon-based fillers on electrical and mechanical properties of a PC/ABS blend. Polymers 2019, 12(1), 29. https://doi.org/10.3390/polym12010029
- Sanya, O.T.; Oji, B.; Owoeye, S.S.; Egbochie, E.J. Influence of particle size and particle loading on mechanical properties of silicon carbide–reinforced epoxy composites. The International Journal of Advanced Manufacturing Technology 2019, 103(9), 4787-4794. https://doi.org/10.1007/s00170-019-04009-1
- Melo, P.M.A.; Macêdo, O.B.; Barbosa, G.P.; Ueki, M.M.; Silva, L.B. High-density polyethylene/mollusk shell-waste composites: effects of particle size and coupling agent on morphology, mechanical and thermal properties. Journal of Materials Research and Technology 2019, 8(2), 1915-1925. https://doi.org/10.1007/s00170-019-04009-1
- Senthil Muthu Kumar, T.; Senthilkumar, K.; Chandrasekar, M.; Subramaniam, S.; Mavinkere Rangappa, S.; Siengchin, S.; Rajini, N. Influence of fillers on the thermal and mechanical properties of biocomposites: an overview. Biofibers and Biopolymers for Biocomposites 2020, 111-133. https://doi.org/10.1007/978-3-030-40301-0_5
- Rani, G.E.; Murugeswari, R.; Siengchin, S.; Rajini, N.; Kumar, M.A. Quantitative assessment of particle dispersion in polymeric composites and its effect on mechanical properties. Journal of Materials Research and Technology 2022, 19, 1836-1845 https://doi.org/10.1016/j.jmrt.2022.05.147
- Samal, S. Effect of shape and size of filler particle on the aggregation and sedimentation behavior of the polymer composite. Powder Technology 2020, 366, 43-51. https://doi.org/10.1016/j.powtec.2020.02.054
References have been renumbered accordingly in the paper.
##Comments on Introduction
- Line 36-37: In the majority of studies, there is no quantitative assessment of the agglomerates, including their number, average size, and concentration. Please add references.
Introduction is modified based on the valuable reviewer’s feedback as follows:
Lines 47 – 54: “For example, paper [10] analyzes how thickness of polymer coating on the surface of reinforcement particles impacts mechanical performance of a composite. Papers [12-16] estimate an impact of distributed particles concentration on mechanical performance of various composites. However, there is no data on the proportion of these particles’ agglomerates in the composite. Papers [17, 18] should be mentioned as well. They study the effect of agglomerates on mechanical performance of polymer composites; however, they do not discuss agglomeration mechanism.”
Lines 73 – 79: “It is possible to determine agglomerates parameters over the entire volume of a polymer composite using optical microscopy, for example. This method provides a possibility to determine the dimensions of large agglomerates (1 micron and up) in optically transparent polymers. That is why this paper uses polymethyl methacrylate as a transparent matrix. Al2O3 submicron particles are originally spheres [9, 10, 24], and polystyrene coating changes their geometry. Due to this fact electron microscopy makes it possible to see if the coating is present on the particles.”
- Please discuss the novelty of the findings compared to reference 10.
A discussion of the novelty was added to the Introduction compared to reference [10] based on the valuable reviewer’s feedback (lines 47 - 51):
“For example, paper [10] analyzes how thickness of polymer coating on the surface of reinforcement particles impacts mechanical performance of a composite. Papers [12-16] estimate an impact of distributed particles concentration on mechanical performance of various composites. However, there is no data on the proportion of these particles’ agglomerates in the composite.”
- Why did the authors choose MMA and Al2O3 for this study? Please include the clarification in the introduction section.
A rationale for using PMMA and Al2O3 is added to Introduction (lines 73 - 79):
“It is possible to determine agglomerates parameters over the entire volume of a polymer composite using optical microscopy, for example. This method provides a possibility to determine the dimensions of large agglomerates (1 micron and up) in optically transparent polymers. That is why this paper uses polymethyl methacrylate as a transparent matrix. Al2O3 submicron particles are originally spheres [9, 10, 24], and polystyrene coating changes their geometry. Due to this fact electron microscopy makes it possible to see if the coating is present on the particles.”
- Are there other reports using the same concept besides Al2O3 and MMA?
The authors are unaware of any publications with the results of studying the parameters of Al2O3 particles agglomerates coated by polystyrene in polymethyl methacrylate polymer.
##Comments on methodology
- The materials section should be separated and outline the details of the materials used in this study.
A subsection called “Materials and their properties” is arranged based on the reviewer’s feedback. And a subsection called “Methods” is arranged as well.
- Please add a polystyrene-coated Al2O3 and composite preparation section and include the flowing clarification.
- What was the density of styrene vapor in the reaction chamber (2), and what was the gas pressure in the chamber (2)?
- What was the turbidity and pH of the water? How have authors used UV irradiation?
- Please provide a complete schematic of this synthesis process.
The authors decided to give a combined reply to the reviewer’s comments No. 10-13.
The authors referenced their papers [9, 10, 24]. Paper [9] discusses preparation of polystyrene-coated Al2O3. The authors are grateful to the reviewer for this valuable feedback and suggest not to overload the article with the coating scheme that has already been published. To make it more convenient for the reviewer, a section of paper [9] with detailed description of the method is presented below (see the attached file).
- Line 74: 'The coating on the surface of the particles was controlled by scanning electron microscopy' How?
The authors referenced their paper [9], where this issue was studied thoroughly. A part of this article is presented below for the reviewer’s convenience (see the attached file).
- With scale, SEM micrographs are meaningful.
The scale is added to the micrographs (Fig. 1 and Fig. 3).
- Line 78: 'A study of adhesion between polystyrene coating and aluminum oxide particles [9]. The reviewer can not access the paper; sorry for it. The reviewer wants to know the mechanism of the formation of covalent bonds between aluminum oxide particles and styrene molecules.
The authors referenced their paper [9], where this issue was studied thoroughly. The corresponding part of this paper is included in the reply to comment 14.
- Lines 82-83: 'Coated particles were deposited on polymethyl methacrylate (PMMA) powder while stirring to get uniform distribution of coated particles in PMMA' How?
The text of the paper was revised based on the reviewer’s feedback (lines 90 – 93):
“For that purpose, a powder of coated particles was gradually added to PMMA powder while stirring using ITA-07 powder mixer (Italy). Mixing speed was 20±5 rpm. After mixing, the specimens of PMMA composite were obtained within 30±5 minutes.”
- Lines 86-87: Please remove the hyphen from each condition, e.g., temperature – 200±5оС.
The hyphens were removed from each condition.
- Please include the image of each specimen in table 1 or Fig. 1.
The authors believe it unreasonable to include the image of each specimen of coated particles in the paper. Size distribution of the particles and a minor number of particles in micrographs do not allow the reader to see any difference in coating thickness. The difference in coating thickness may be determined only based on the results of statistical analysis of the micrographs. A method of micrographs analysis is presented in paper [9]. For the reviewer’s convenience, large parts of this paper are included in the replies to comments 13 and 14.
- Please separate the characterization section.
The following sections are arranged in the paper based on the reviewer’s feedback:
2.1. Materials and their properties.
2.2. Methods
- Results
- Discussion
- Lines 95-96: What type of micrographs were used for analyzing agglomerates by ImageJ software? Are these SEM images? What was the scale? Have authors used the same scale images for each specimen? Please provide those micrographs in a figure.
The paper (line 95) says that optical microscopy was used to analyze agglomerates. “Analysis of the agglomerates” subsection (lines 101-103) says: Micromed microscope with FMA050 Touptek Photonics adapter with 40X magnification made it possible to measure approx. 1 ± 0.1 micron agglomerates.
An example of a micrograph is presented below and is included in the paper (Fig. 4).
- Line 100: Please provide the histograms in Figure 2.
The authors provided the histogram in Fig. 5.
Figure 5 shows an example of agglomerates size distribution histogram for specimen 2.2. Figure 4 demonstrates an example of a micrograph for specimen 2.2.
- Therefore, Lines 58-112; need modifications.
Lines 58-112 were modified.
##Comments on results and discussion
- Please delete the comma and use a point in the scale of Fig. 2b.
Commas are changed to points in Fig. 2b.
- The scale of magnification must be included in Fig. 3
The scale is included in Fig. 3.
- Lines 134-139: It is in brief. Please a detail.
In lines 134-139 the authors speculate about agglomeration mechanisms. The authors did not aim at evaluating the concentration effect on the parameters of elastic and inelastic collisions of the particles in polymer. We agree with the reviewer that these studies are by far of interest and may be the subject of a separate article. The assumptions about agglomeration mechanisms, made by the authors, are based on the well-known principles of particles collision dynamics [13]. There is no point in theoretical estimation of elastic and inelastic collisions cross-section due to the lack of data about the impact factors.
- If possible, please add at least two characteristics of composites from FTIR, XRD, and TGA.
The reviewer is right: these characteristics are required to determine the structure and mechanical properties of polymers. However, the authors did not aim at studying the properties of polymer composites that contain reinforcement particles and their agglomerates. That is why these characteristics are not mentioned in the paper.
- Please add a discussion section separately.
This section is added separately based on the reviewer’s feedback (lines 170 – 213).
##Comments on conclusion
- Some of the results and discussion are in the conclusion section. Please write the findings and limitations of this research, future scope, and utility.
The authors revised Conclusion section based on the reviewer’s feedback.
Lines 218 – 228: “Only large-scale agglomerates (exceeding 1 micron) were analyzed, so optical microscopy could be used to get the images of agglomerates in optically transparent PMMA specimens and statistical analysis of these images could be performed. Unlike uncoated aluminum oxide particles, when coated aluminum oxide particles are used, agglomerates concentration in polymer composites decreases approx. 10 times. Moreover, when particles concentration in the specimens increases, the ratio of agglomerates concentration in the specimens reinforced with the original particles and in the specimens with coated particles increases as well. It is demonstrated that modification of submicron particles by polymer coating reduces the number of agglomerates in a polymer composite. The paper highlights that parameters of the agglomerates depend on the frequency of inelastic collisions of distributed particles in a polymer melt during composite specimens molding.”
Lines 239 – 246: “When the concentration exceeds 0.2%, transmittance values for the specimens reinforced with coated particles are greater than the ones for the specimens reinforced with the original particles. When the concentration is below 0.2%, the size of agglomerates in the specimens reinforced with coated particles is approx. twice as large as in the specimens reinforced with the original particles. A suggested method for determining the parameters of submicron particles agglomerates in optically transparent polymers (polymethyl methacrylate, polycarbonate) may be applied for various particles and surface modification techniques.”

Round 2
Reviewer 2 Report
Dear authors, Thank you for revising the manuscript in response to the reviewer's comments. Although some comments are not resolved, the overall response is acceptable. Therefore, I am now recommending that the manuscript be published.
Minor comments on proof corrections.
1. Line 51: 'Papers [17,18] should be mentioned as well'. Please check the meaning. Lines 52-53: Please check English.
2. One request is that you adjust the image's background color in Fig. 4 if the particles are visible with different light colors. And please remove the square box from Fig. 5.
Thank you so much, and best wishes!